# Magnetic-Activated Cell Sorting as a Method to Improve Necrozoospermia-Related Asthenozoospermic Samples

**DOI:** 10.3390/jcm11102914

**Published:** 2022-05-21

**Authors:** Gábor Máté, András Balló, László Márk, Péter Czétány, Árpád Szántó, Attila Török

**Affiliations:** 1Pannon Reproduction Institute, 8300 Tapolca, Hungary; ballo.andras@pte.hu (A.B.); drtoroka@t-online.hu (A.T.); 2Urology Clinic, University of Pécs Clinical Centre, 7621 Pécs, Hungary; czetany.peter@pte.hu (P.C.); szanto.arpad@pte.hu (Á.S.); 3National Human Reproduction Laboratory, University of Pécs, 7624 Pécs, Hungary; laszlo.mark@aok.pte.hu; 4Department of Analytical Biochemistry, Institute of Biochemistry and Medical Chemistry, University of Pécs Medical School, 7624 Pécs, Hungary; 5MTA-PTE Human Reproduction Research Group, University of Pécs, 7624 Pécs, Hungary

**Keywords:** assisted reproduction, asthenozoospermia, DNA fragmentation, in vitro fertilization, magnetic-activated cell sorting, necrozoospermia, vitality

## Abstract

According to some statistics, absolute asthenozoospermia affects every 1 in 5000 men. Although this incidence rate does not appear to be too high, it is extremely important to address the phenomenon because it can drastically reduce the chances of pregnancy, even with assisted reproduction. The biggest problem with absolute asthenozoospermia is that it is difficult to distinguish between live and dead sperm cells, and fertilization with non-viable spermatozoa may contribute to the failure of an assisted reproduction cycle. Nowadays, DNA fragmentation (DF) is a crucial parameter of semen analysis, and in this paper, we provide evidence of the correlation between DF and vitality. For this purpose, the main semen parameters were investigated by a CASA system (concentration, motility, progressive motility, vitality and DF). In the necrozoospermic group (vitality < 58%), all the measured parameters showed significant differences compared to normal vitality. Concentration (30.1 M mL^−1^ vs. 13.6 M mL^−1^), motility (31.9% vs. 18.3%), and progressive motility (24.3% vs. 12.7%) were significantly decreased, while DF was significantly increased (17.4% vs. 23.7%). Based on the connection between vitality decrement and DF increment, DF lowering methods, such as magnetic-activated cell sorting, have been hypothesized as novel methods for the elimination of dead spermatozoa.

## 1. Introduction

Infertility is a growing problem; it affects 10–15% of couples worldwide, and half of the cases are related to the male partner alone or coupled with the female partner. With the intracytoplasmic sperm injection (ICSI) technique, most of these male-related problems can be compensated, whether it is oligozoospermia, teratozoospermia or asthenozoospermia. However, in the case of absolute asthenozoospermia, when no motile sperm are found in the used sample, it is very hard to distinguish between viable and dead sperms during ICSI, which reduces its success. The nature of absolute asthenozoospermia is still not fully understood, but a lot of research has been conducted in this regard. Potential triggers include metabolic deficiencies of spermatozoa, ultrastructural disorders of sperm flagellum, necrozoospermia, and it can also be idiopathic [1]. The origin of dead spermatozoa can be a result of either apoptotic or necrotic processes. A lot of dyes have been developed to determine the characteristic of these viable/apoptotic/necrotic cells, but a great disadvantage of these techniques is that they are only diagnostic methods, and the fertilization process is not improved by them. The hypoosmotic swelling test (HOST) may be the only available technique that allows us to make correlations between the viability of cells and the integrity of their plasma membrane [2]. Besides this, a novel technique in sperm separation is magnetic-activated cell sorting (MACS), which decreases the DNA fragmentation (DF) of sperm samples. Its function is based on the fact that in apoptotic cells, the phosphatidylserine molecule is transferred from the inner surface of the plasma membrane to the outer surface, which enables it to be labelled and separated in a magnetic field using magnetically coupled annexin V [3]. MACS is generally applied to reduce DF, but our study aimed to present a lesser known ability of MACS separation, namely, how to enhance the amount of viable, but immotile, sperms in ICSI samples. For this purpose, we will demonstrate (i) the correlation between DF and sperm vitality, and (ii) the effects of MACS separation on vitality through a case study.

## 2. Materials and Methods

Semen samples were collected from 205 men who applied for sperm DF determination in the Hungarian Pannon Reproduction Institute in 2021. Samples were divided into two groups, based on the percentage of viable spermatozoa or the value of DF. In the first case, the first group contained 151 samples with sperm vitality higher than 58%, and the second group contained 54 samples with necrozoospermic values (vitality was lower than 58%) [2]. In the second case, the low DF group (<30%) had 170 samples and the high DF group (>30%) had 35 samples. Semen analysis was performed by an SCA SCOPE (Microptic S.L., Barcelona, Spain) automatic semen analysis system, and the main parameters were measured, namely, sperm vitality, DF, concentration, motility and progressive motility. Samples were collected after 3 days of sexual abstinence, and assessments were performed after liquefaction at room temperature, but not more than 60 min after ejaculation. Sperm vitality was assessed after eosin staining. One drop of semen sample (5 µL) was mixed with equal volume of 1% eosin-Y solution, a smear was made on a glass side, covered with a coverslip, incubated for 30 s at 37 °C, and 200 spermatozoa were evaluated. DF was assessed using the sperm chromatin dispersion (SCD) test, following the manufacturer’s instructions and the WHO guidelines [2]. Briefly, semen was diluted with phosphate buffer saline to 5–10 million mL^−1^. A 30 µL sample was mixed with 1% agarose at 37 °C. A total of 14 μL of the above mixture was pipetted onto a prepared agarose glass slide and covered with a glass coverslip (18 × 18 mm). The slide was cooled for 5 min at 4 °C. The coverslip was removed and the glass slide was immersed horizontally into a denaturation solution (0.08 M HCl) for 7 min at room temperature. After this, the slide was transferred to a lysis buffer (0.4 M Tris, 0.4 M DTT, 50 mM EDTA, 0.3% SDS and 1% Triton Xysis) and incubated for 25 min at room temperature. The slide was thoroughly washed with distilled water, dehydrated for 2 min in each of 70, 90 and 100% ethanol and subsequently air dried. The slide was stained with Giemsa and analyzed by an SCA Scope system. During the analysis, halos around the head of sperm cells were observed. Healthy cells show extended or moderate halos, while sperms with fragmented DNA show small or no halos. A minimum of 200 spermatozoa per sample were evaluated. For SCD tests, all of the mentioned reagents were purchased from Microptic S.L as kits.

During the analysis of these 205 samples, absolute asthenozoospermia was observed in one case, and MACS separation was applied for further investigation. For MACS (Miltenyi Biotec, Bergisch Gladbach, Germany) separation, after liquefaction, density gradient centrifugation was performed at 400 g for 12 min. The pellet was collected and washed with a binding buffer. The suspension was centrifuged at 400 g for 3 min and the supernatant was removed. Sperm cells were labelled with 100 µL of magnetic bead-conjugated annexin V for 15 min at room temperature. A separation column containing iron balls was placed in a magnet, and the labelled cells were separated by the magnetic field, allowing the negative fraction to be collected. Annexin V-positive apoptotic sperm cells with fragmented DNA are retained by the column. Washing was performed on the negative fraction, and the sperm cells were then analyzed. For MACS separation, all of the mentioned reagents were purchased from Miltenyi Biotec as kits. Based on the abovementioned methods, DF index and vitality were determined again after MACS separation.

Data were given as average and deviation. A Shapiro–Wilk test was used to evaluate the distribution of the data. Non-normally distributed variables were examined using the Kruskal–Wallis non-parametric test. Graphpad InStat software, version 7.0 (Graphpad Software, San Diego, CA, USA) was used for statistical analysis.

## 3. Results

Both the decrement in sperm vitality and the increment in DF resulted in significant differences (Figure 1). If our results are analyzed as a function of vitality, the DF was 1.36-fold higher in the necrozoospermic group (17.38 ± 10.38% vs. 23.71 ± 16.17%, *p* = 0.0261) in comparison with normal vitality. The concentration, the motility and the progressive motility were 30.07 ± 29.50 M mL^−1^ vs. 13.58 ± 16.81 M mL^−1^, 31.86 ± 16.65% vs. 18.31 ± 11.94%, and 24.31 ± 14.11% vs. 12.67 ± 9.95%, respectively (Figure 1A). If our data are approached from the function of DF, a more significant difference can be obtained (*p* < 0.001). In the low DF group, the vitality was 66.91 ± 14.08%, and in the high DF group, this value was 56.25 ± 16.44%. The concentration, the motility and the progressive motility were 26.52 ± 28.51 M mL^−1^ vs. 21.85 ± 23.22 M mL^−1^, 30.51 ± 16.67% vs. 17.51 ± 11.53%, and 23.01 ± 14.23% vs. 12.68 ± 9.68%, respectively (Figure 1B).

In our case study, the raw semen sample contained 8 M mL^−1^ spermatozoa with 0% motility. Cells had 10% vitality and 77.6% DF. After MACS separation, the vitality of spermatozoa increased from 10% to 73% (Figure 2). The DF index of separated sperm cells showed similar improvement, namely, the DF index decreased from 77.6% to 28.2% (Figure 3).

## 4. Discussion

Severe or absolute asthenozoospermia, without any motile sperm, is a problem that makes the fertilization process much harder because the distinction between viable and dead sperm cells is circumstantial. Only a few techniques are available for this purpose, for example, HOST, mechanical touch or laser-assisted selection of spermatozoa, but their application and success are still controversial [1]. In these cases, testicular sperm extraction (TESE) is recommended to isolate live spermatozoa with lower DF; however, this invasive process is often refused by the patient [4]. In our study, we presented evidence of how MACS separation is capable of improving sperm vitality in absolute asthenozoospermic samples. MACS is generally identified as a DF reducing method, but several correlations can be found between vitality and DNA integrity in the literature. Brahem et al. [5] observed vitality-dependent alterations in the DF index, namely, the lower the vitality, the higher the DF. In addition, Samplaski et al. [6] found <30% DF when the vitality was over 75% for most of the samples. This phenomenon has also been confirmed by others [7]. In agreement with the literature, a positive correlation was found between the severity of vitality/necrozoospermia and sperm DF (Figure 1). Both the decrement in sperm vitality and the increment in DF resulted in significant differences (Figure 1). Vitality had a significant impact on all of the measured parameters; DF showed significant increment, and concentration, motility, and progressive motility showed decrement. Increasing the DF resulted in similar effects; only the concentration did not change significantly. However, DF has a more significant impact on vitality than vice versa. Nothing proves this connection better than the significant improvement in the DF index after swim-up sperm preparation. During swim-up, the motile fraction is separated from the immotile fraction, resulting in a strong increment in vitality, and due to the above findings, decrement in the DF [8,9,10,11]. This phenomenon is exploited by microfluidic separation chips, which improve the DF index, motility and, hence, vitality [12]. Unfortunately, the method is not suitable for immotile spermatozoa-containing semen samples. In another study, progressive motility was affected by altered DF; however, vitality was not influenced in this case [13]. Another very important conclusion can be drawn from the above statements, namely, some of the results of DF tests may be misleading because the results also include the data of non-living sperm cells. DF tests provide information on natural fertility, but these results could be incorrect if associated with the effects on the outcome of IUI/IVF/ICSI cycles. For IUI/IVF/ICSI cycles, one of the sperm preparation methods is used (density gradient centrifugation/swim-up), and in these cycles, the applied preparation method itself can reduce DF and improve vitality/motility [8,9,10,11]. Due to this, the results of DF tests and the outcomes of IVF/ICSI cycles cannot clearly determine the effect of fragmentation on clinical pregnancy, miscarriage or live birth rates; both neutral and negative effects can be found. To date, the effect is unclear; several meta-analyses are needed [14].

As we have discovered that there is a strong connection between vitality and DF, a fragmentation reducing method could be suitable for vitality improvement. MACS separation has been found to be appropriate for this purpose. During MACS sample preparation, cells are labelled with annexin V, which allows the molecules to bind with phosphatidylserine in the plasma membrane (Said et al., 2006). This is usually located in the inner layer of the plasma membrane, but during apoptosis, it moves to the outer surface and becomes accessible to the annexin V. On the other hand, in some necrotic cells, phosphatidylserine will be accessible on the inner surface due to the altered state of the plasma membrane (Figure 4). Using this approach, relevant increases in vitality were experienced after annexin V separation occurred (Figure 2). With MACS separation, the vitality of spermatozoa improved 7.3-fold (Figure 2). Although vitality did not rise to 100% and the separated sperm cell was still motionless, this method statistically improves the chances of fertilization with living spermatozoa. As the beneficial effects of MACS on DF have been published several times [15,16,17,18], our results are in agreement with the literature, showing huge decrements in the DF index (77.6% vs. 28.2%, Figure 3). Martínez et al. [19] discussed how MACS could not reduce DF in a small subpopulation of sperm samples, but they noticed its significant reducing capability in samples with highly degraded DNA. This observation also supports our results.

## 5. Conclusions

In conclusion, our results and the literature clearly demonstrate the negative effects of DF on vitality and vice versa. Using this close relationship between the two parameters, we hypothesized that a method to reduce fragmentation could improve vitality. In addition, we were looking for a method that is suitable for the preparation of absolute asthenozoospermic samples; its operation is not based on sperm motility. The MACS technique has been found for this purpose. Annexin V is capable of binding both early apoptotic and late apoptotic/secondary necrotic cells, inducing the increment in the ratio of viable cells after magnetic separation. Both this study and the literature data suggest that the DF measurements of semen samples essentially reflect the levels of dead/dying sperm cells. The DF is, therefore, decreased by the elimination of these cells during most of the sperm preparation methods. The great limitation of this study is that it is only a case study, and the case number was not high enough due to the occurrence of absolute asthenozoospermia; therefore, no significant conclusions can be drawn. Further investigations are required, but our principle could be useful for TESE samples where no motile spermatozoa are found. Another problematic issue could be the potential cost of the method. The price can vary from center to center and manufacturer to manufacturer, but the average material cost of EUR 200–300 is far lower than the price of a fresh cycle. As this is a relatively cost-effective method, it should not limit its widespread use for further studies.

## Figures and Tables

**Figure 1 jcm-11-02914-f001:**
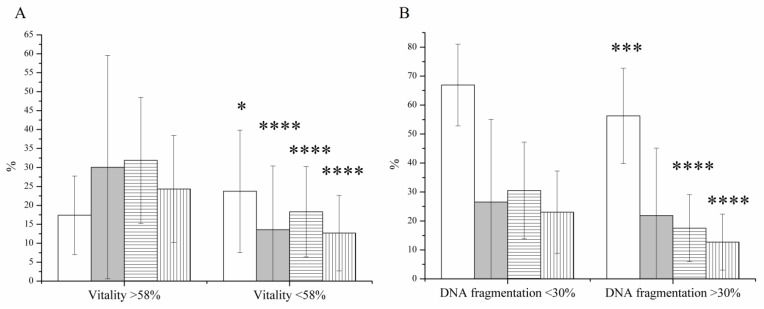
Correlation between sperm vitality and their DNA fragmentation. (**A**) DNA fragmentation (white column), concentration (grey column), motility (horizontally striped column) and progressive motility (vertically striped column) as a function of sperm vitality. (**B**) Vitality (white column), concentration (grey column), motility (horizontally striped column) and progressive motility (vertically striped column) as a function of DNA fragmentation. * *p* < 0.05, *** *p* < 0.001, **** *p* < 0.0001.

**Figure 2 jcm-11-02914-f002:**
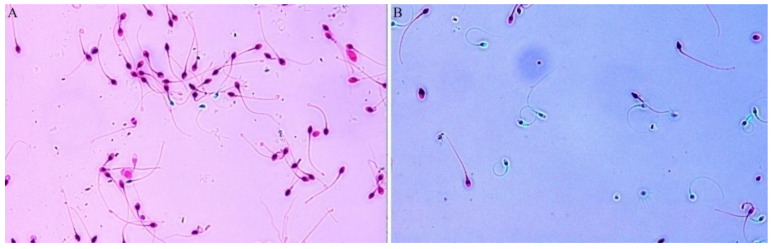
Sperm vitality (**A**) before and (**B**) after magnetic-activated cell sorting. Cells were labelled with eosin dye. Dead spermatozoa became red and viable spermatozoa remained colorless.

**Figure 3 jcm-11-02914-f003:**
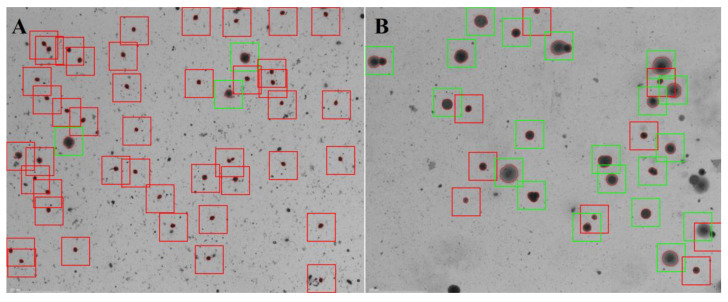
Demonstrative figure of sperm DF (**A**) before and (**B**) after magnetic-activated cell sorting. Cells were labelled based on the SCD method. Healthy cells show extended or moderate halos (green squares), while sperms with fragmented DNA show small or no halos (red squares).

**Figure 4 jcm-11-02914-f004:**
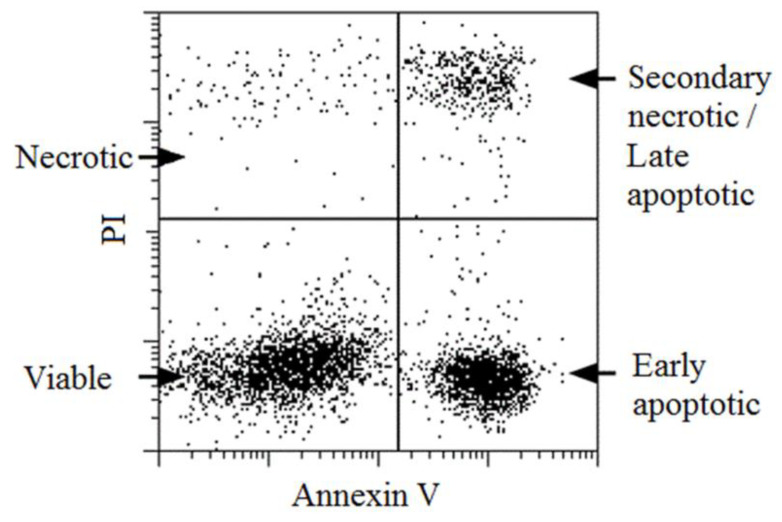
Schematic figure of annexin V positivity of sperm cells. Dead cells can be labelled basically with propidium iodide, but in some cases, these cells are also positive for annexin V.

## Data Availability

Not applicable.

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
