# Peer review of "Magnetic-Activated Cell Sorting as a Method to Improve Necrozoospermia-Related Asthenozoospermic Samples"

_jcm, 2022, doi:10.3390/jcm11102914_

Round 1

Reviewer 1 Report

I would appreciated to see also figure with results of sperm chromatin dispersion test (DSS test), why authors didnt show this results, but only sperm vitality staining by eosin?

Author Response

Thank you for your comment! We have tried to focus attention on the effects of separation on sperm vitality because several articles have been already published about SCD tests and MACS separations. As the Reviewer suggested, a figure of SCD test results has been added to the manuscript.

Reviewer 2 Report

The article show a new method for sperm selection in acinellature spermatozoa. 
the idea it’s fine! There is a lack about the  cost of the procedures that in my opinion could de the big problem of the dagli use of the technique. 
could be add and detailed? This could give a better push to use or not the 

Author Response

Thank you for your comment! In comparison with the cost a standard ICSI cycle (stimulation, oocyte pick-up, lab work, ICSI, etc.), the cost of this technique is much lower. Its cost is around 200-300 euro (depending on the country or the manufacturer).  As the Reviewer suggested, a few sentences about the potential cost has been added to the “Conclusion” section.

Reviewer 3 Report

The article entitled "Magnetic-activated cell sorting as a method to improve necrozoospermia-related asthenozoospermic samples" aimed to verify the ability of MACS separation in necrozoospermic samples. 

The idea of developing a new tool to improve the separation of better spermatozoa for IVF is of great importance, once clinicians are always looking for better results. But the main question that remains here is the usefulness of this technique during the IVF process. After incubation and MACS separation of the spermatozoa, you cannot use it for ICSI, once you applied an antibody to the cell. So, therefore, how the authors would solve this problem? The use of MACS in research is very wide, helpful, and not new, once you can separate some populations of interest is a small period of time. 

Another problem of the article is MACS effect on DNA fragmentation evaluation. The authors said that this was published several times, and cited reference 15 (only one). In addition, this parameter should be evaluated in your project, once the idea here is to standardize the results. In this line, why SCD was chosen? This is the test of the worst quality to evaluate DNA fragmentation. Also, the methods section should be described better, such as solutions concentrations and reagents.

Author Response

Remarks: The idea of developing a new tool to improve the separation of better spermatozoa for IVF is of great importance, once clinicians are always looking for better results. But the main question that remains here is the usefulness of this technique during the IVF process. After incubation and MACS separation of the spermatozoa, you cannot use it for ICSI, once you applied an antibody to the cell. So, therefore, how the authors would solve this problem? The use of MACS in research is very wide, helpful, and not new, once you can separate some populations of interest is a small period of time.

Answers: Thank you for your comment! The used MACS annexin-V kit is suitable for clinical use. During annexin-V labelling, the antibody bounds to the apoptotic, fragmented cells. Since the antibody is coupled with a magnetic bead, after separation the antibody and the labelled cells are retained on the separation column. For ICSI, only the antibody-negative fraction is used. Therefore, the oocytes do not get into contact with it.

Remarks: Another problem of the article is MACS effect on DNA fragmentation evaluation. The authors said that this was published several times, and cited reference 15 (only one). In addition, this parameter should be evaluated in your project, once the idea here is to standardize the results. In this line, why SCD was chosen? This is the test of the worst quality to evaluate DNA fragmentation. Also, the methods section should be described better, such as solutions concentrations and reagents.

Answers: Thank you for your comment! The Reviewer is right, only one reference was cited. Additional references have been cited in the text and a demonstrative figure has been added about the result of SCD test. SCD test is one of the accepted DNA-fragmentation measurements. The newest WHO guideline also mentions it as an accepted method. There was no professional argument behind its use. Unfortunately, this is the only method available in our institute. As the Reviewer suggested, the methods section has been described better.

Round 2

Reviewer 3 Report

The authors accepted and improved the manuscript as suggested, and now is ready for publication.